# Comparative Investigation of XPS Spectra of Oxidated Carbon Nanotubes and Graphene

**Viktor P. Afanas'ev** [1], **Grigorii S. Bocharov** [1], **Alexander V. Eletskii** [1,*], **Lidiya G. Lobanova** [1], **Konstantin I. Maslakov** [2] **and Serguei V. Savilov** [2]

[1] Institute of Thermal and Nuclear Power Engineering, National Research University MPEI, 111250 Moscow, Russia; v.af@mail.ru (V.P.A.); bocharovgs@mail.ru (G.S.B.); lida.lobanova.2017@mail.ru (L.G.L.)

[2] Department of Chemistry, Lomonosov Moscow State University, 119991 Moscow, Russia; nonvitas@gmail.com (K.I.M.); savilov@mail.ru (S.V.S.)

[*] Correspondence: eletskii@mail.ru; Tel.: +8-916-905-36-37

**Abstract:** X-ray photoelectron emission spectra of thermally reduced graphene oxide samples and carbon nanotubes (CNTs) with various oxidation degrees are presented in this paper. A method for the reconstruction of differential electron inelastic scattering cross sections from the energy loss spectra of photoelectrons is described and discussed. The analysis of the part of the characteristic photoelectron energy loss spectrum adjacent to the C1 peak indicated a considerable influence of the thermal reduction of graphene oxide on the electron properties of the samples obtained. On the contrary, the oxidation of CNTs by refluxing in a concentrated $HNO_3$ solution does not change the free electron excitation spectrum.

**Keywords:** graphene oxide; thermal reduction; carbon nanotube oxidation; X-ray photoelectron spectroscopy (XPS); photoelectron spectra analysis (PES analysis); allotrope carbon modifications; differential inverse inelastic mean free path (DIIMFP)

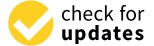



## 1. Introduction

The oxidation of carbon nanomaterials changes their electron properties considerably [1,2]. Thus, graphene oxide shows limited conduction of electricity, while its thermal reduction results in an enhancement of the conductivity, similar to the reference value for graphite at an annealing temperature of approximately 800 °C [3–5]. The most informative source of data on the electronic characteristics of a material is X-ray photoelectron spectroscopy (XPS), which does not inflict damages on the material [6]. Particularly, the results of XPS spectra evolution on the thermal reduction of graphene oxide permitted the determination of interconnection between the oxidation degree of the material and the intensity of plasmon oscillations, which in its turn relates to the free electron concentration [4,5,7,8].

The electron characteristics of carbon nanotubes (CNTs) also depend on the oxidation degree of samples. This relationship is studied in the present work on the basis of the treatment of XPS spectra of samples obtained on CNT oxidation. These spectra are compared with those for reduced graphene oxide with various oxidation degrees. The comparison indicates a considerable difference in the electronic characteristics of these two nanocarbon modifications.

The main information on the electronic characteristics of the sample under investigation are the position and intensity of peaks formed by photoelectrons escaping in vacuum without energy loss (peak shape analysis—PSA) [6]. In addition, an approach based on the XPS spectra in the characteristic-energy-loss region adjacent to the peaks (photoelectron spectra analysis—PES analysis) was used.

Samples of thermally reduced graphene oxide and oxidized CNTs were studied by processing XPS spectra of multiple inelastic electron energy losses. Various approaches have

been developed to derive the differential inverse inelastic mean free path (DIIMFP–$\omega_{in}(\Delta)$) from the energy spectrum of photoelectrons; $\Delta$—energy loss. A comparison of the $\omega_{in}(\Delta)$ of graphene oxide and CNTs indicates a considerable difference in the behavior of their $\pi$ bonds. Thus, the oxidation of CNTs does not break the $\pi$ bonds. The $\omega_{in}(\Delta)$ for graphene oxide annealed at a temperature of 600–900 °C corresponds to that of pyrolytic graphite.

The electrons escaping in vacuum without energy loss do not interact with the electrons of the sample and do not carry information on the electron structure of the sample. This information is contained in the $\omega_{in}(\Delta)$, which determine the energy spectrum of characteristic electron excitations and permit determination uniquely the type of carbon allotrope modification. Figure 1 presents the $\omega_{in}(\Delta)$ for various allotrope carbon modifications [9].

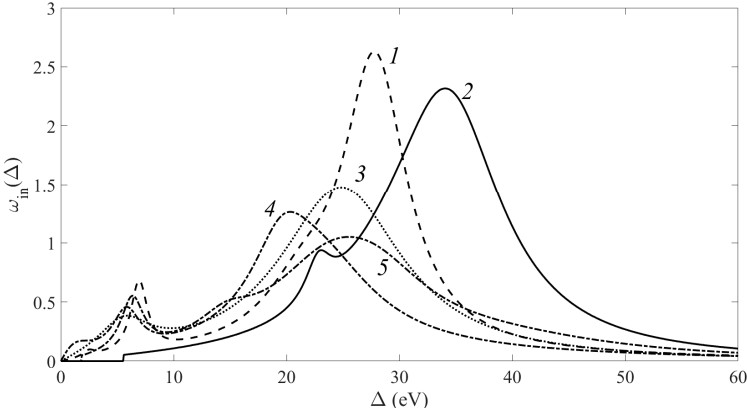

**Figure 1.** The $\omega_{in}(\Delta)$ calculated for various allotrope carbon modifications: 1—graphite, 2—diamond, 3—amorphous carbon, 4—glassy, and 5—$C_{60}$ [9].

As has been shown by Tougaard [10], PSA approach provides rather ambiguous results for the analysis of non-homogeneous materials. However, it is convenience as there are a great quantity of textbooks [6] and program codes permitting one to quickly obtain information on the sample under study. The code Casa XPS [11] is but one example. One should note that the abbreviation XPS is presently used for labeling PSA, while the exploration of samples on the basis of the $\omega_{in}(\Delta)$ derived from XPS spectra taking into account multiple electron energy losses is called PES analysis.

## 2. Experiment

Graphene oxide (GO) was prepared from graphite oxide synthesized by the standard Hummers method [12]. The material presents a multi-layer paper-like film of 40–60 µm in thickness, consisting of a large number of GO fragments of several µm in size. The film was cut for samples of 30 µm × 15 µm in size, which were subjected to thermal treatment and subsequent investigation. The thermal treatment of samples was performed using a high temperature furnace, Planar GROW-2S. The samples were placed into a quartz boat of 20 cm × 3 cm × 2.5 cm in size, which was inserted into the furnace camera. The thermal treatment of samples was performed under a slow flow of Ar at a velocity of 50 cm$^3$/min (reduced to the normal conditions) and a pressure of up to 10 Torr.

The XPS spectra of GO samples annealed at various temperatures (25–1000 °C) were measured by means of the X-ray source MX650 using the monochromatic line Al K$\alpha$. The spectra were registered by means of a spherical energy analyzer R4000 (both devices of Scienta-Omicron company).

Figure 2 presents the C1 spectra of samples that were GO annealed at temperatures of 150 °C and 600 °C. The range of the annealing temperature was extended for the second stage of experiments.

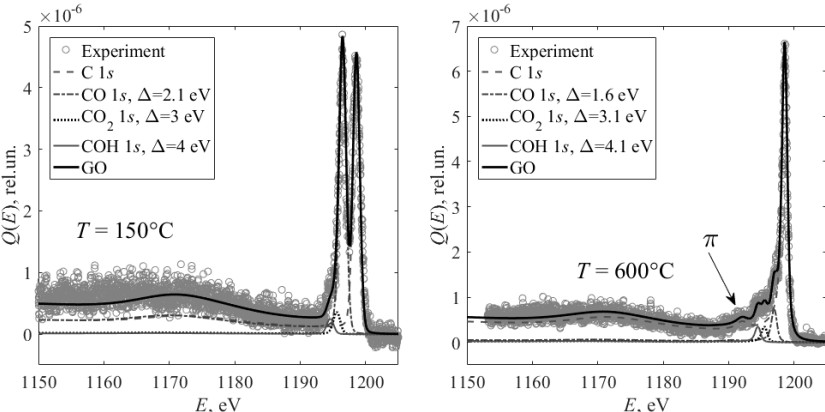

**Figure 2.** C1 spectra of GO samples annealed at various temperatures: left—150°; right—600°. Circles—experiment for GO; solid dark line—calculation for GO (Equation (1)); dashed line—C1s; dashed-dotted line—CO1s; dotted line—CO$_2$1s; solid light line—COH1s; E—energy of photoelectrons.

Multi-walled CNTs were synthesized at 650 °C using the catalytic chemical vapor deposition (CCVD) method via the technique described in [13,14]. Hexane and Co-Mo/MgO were used as a carbon source and a growth catalyst, respectively. The synthesized material was annealed in air flow at 400 °C to remove amorphous carbon, and then washed with HCl and distilled water to eliminate metal impurities. The obtained sample of pristine CNTs was designated as "CNTp". The oxidation of CNTs was performed in a concentrated solution HNO$_3$ (Chimmed, Moscow, Russia, 99.99%) under intense stirring for 1, 3, 6, 9 and 15 h [15]. The material obtained was filtered and washed by distilled water to reach a neutral pH and then dried at 130 °C. The samples were labeled as "CNTn", where "n" is the duration of stirring. XPS spectra of the C1 peaks were measured using the setup Kratos Axis Ultra DLD.

Figure 3 presents XPS spectra of CNTs for samples with various oxidation degrees. The spectrum of the original sample is shown by a dotted line. As shown, both the position and the shape of the plasmon π-peak do not depend on the oxidation degree.

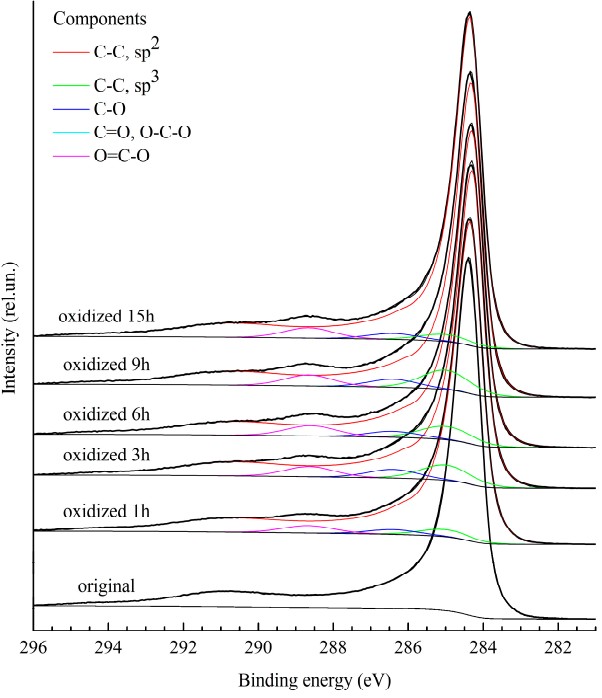

**Figure 3.** XPS spectra of the C1s line measured for CNT samples for different durations of oxidation.

### 3. Analysis

This chapter describes methods for the deconvolution of the single-scattering cross section, which univocally determines allotrope carbon modification, as demonstrated by Figure 1.

The differential density of photoelectron flow $Q(\tau, \Delta, \mu_0, \mu, \varphi)$ is expressed using the representation of partial intensities [16,17]:

$$Q(\tau, \Delta, \mu_0, \mu, \varphi) = Q_0(\tau, \mu_0, \mu, \varphi)\delta(\Delta) + \sum_{k=1}^{\infty} Q_k(\tau, \mu_0, \mu, \varphi)x_{in}^k(\Delta), \tag{1}$$

where $\tau = z/l_{tot}$ is the ratio of the free path of a photoelectron in the target $z$ to the total path $l_{tot}$; $l_{tot}^{-1} = l_{in}^{-1} + l_{el}^{-1}$; $l_{in}$ and $l_{el}$—inelastic and elastic mean free path, correspondingly; $\mu_0$ and $\mu$ are the cosine of the angle of incidence and the angle of scattering, correspondingly; $\theta_0 = \arccos(\mu_0)$ and $\theta = \arccos(\mu)$ are polar angles for electron take in and take off from the normal to the surface; $\varphi$ is the azimuthal angle; $Q_k(\tau, \mu_0, \mu, \varphi)$—partial coefficients or the probability of a photoelectron to lose energy $\Delta$ as a result of $k$ sequential acts of inelastic scattering [16]; $x_{in}^k(\Delta) = \int_0^{\Delta} x_{in}(\Delta - \varepsilon)x_{in}^{k-1}(\varepsilon)d\varepsilon$ is the probability to lose energy $\Delta$ as a result of $k$ successive inelastic scattering.

The representation of the partial intensities (1) permit one to describe both photoelectron (PES) and reflected electron spectra in a common form. This approach is called as reflected electron energy loss spectrometry (REELS). In the first approximation, assume that the normalized differential inverse inelastic mean free path (nDIIMFP)–$x_{in}(\Delta) = \omega_{in}(\Delta)/\sigma_{in}$ is homogeneous over the target ($\sigma_{in}$—inelastic cross section). Deconvolution of $x_{in}(\Delta)$ from REELS data is performed using a modification of the known Tougaard method [18]. The representation (1) makes it possible to use the the deconvolution method for both REELS and PES spectra. The partial coefficients $Q_k(\tau, \mu_0, \mu, \varphi)$ are calculated by the method described in [19–21]. The PES is normalized, dividing it by $Q_0$ (the first term in Equation (1)). The term describing the peak of photoelectrons in vacuum without a loss in energy is removed from Equation (1). This peak differs in its shape from the $\delta$ function due to the influence of the hardware function of the energy analyzer, the Doppler effect and a complicated function describing the spectrum of the formed photoelectrons. The combined effect of the above-listed factors results in the formula by Doniach and Sunjic [22].

The energy loss axis $\Delta$ is broken into intervals with step $h$:

$$\Delta_i = ih. \tag{2}$$

In the low-energy-loss region, the contribution of multiple electron scattering is negligible. Further, it is assumed that:

$$x_{in}(0) = 0 \tag{3}$$

This implies:

$$Q(\Delta_0 = 0) = 0. \tag{4}$$

Hereafter, for the sake of simplification, the term $x(\Delta)$ will be used instead of $x_{in}(\Delta)$. Taking into account approximations (3) and (4), $k = 1$ in Equation (1), one obtaines:

$$Q(\Delta_1) = C_1 x(\Delta_1), \tag{5}$$

where $C_1 = Q_1(\tau, \mu_0, \mu, \varphi)/Q_0(\tau, \mu_0, \mu, \varphi)$.

Supposing further that $x(\Delta) \equiv x_{in}(\Delta)$, one obtains the following relation for $x(\Delta_1)$:

$$x(\Delta_1) = \frac{Q(\Delta_1)}{C_1}. \tag{6}$$

For calculation of $Q(\Delta_2)$, two terms in Equation (1) should be taken into consideration:

$$Q(\Delta_2) = C_1 x(\Delta_2) + C_2 x^2(\Delta_2), \tag{7}$$

where $C_2 = Q_2(\tau, \mu_0, \mu, \varphi)/Q_0(\tau, \mu_0, \mu, \varphi)$.

For calculation of $x^2(\Delta_2)$, the expression $x_{in}^k(\Delta) = \int_0^\Delta x_{in}(\Delta - \varepsilon)x_{in}^{k-1}(\varepsilon)d\varepsilon$ is used. Integration by the rectangle method results in:

$$x^2(\Delta_2) = h(x(\Delta_0)x(\Delta_2) + x(\Delta_1)x(\Delta_1)) = hx(\Delta_1)x(\Delta_1). \tag{8}$$

Substituting (8) into (7), one obtains:

$$Q(\Delta_2) = C_1 x(\Delta_2) + C_2 hx(\Delta_1)x(\Delta_1). \tag{9}$$

Equation (9) permits the determination of $x(\Delta_2)$:

$$x(\Delta_2) = (Q(\Delta_2) - C_2 hx(\Delta_1)x(\Delta_1))\frac{1}{C_1}. \tag{10}$$

The step-by-step movement to the high-energy-loss region results in the following equation for $x(\Delta_k)$:

$$x(\Delta_k) = \left(Q(\Delta_k) - C_2 x^2(\Delta_k)h - C_3 x^3(\Delta_k)h^2 - \cdots - C_k x^k(\Delta_k)h^{k-1}\right)\frac{1}{C_1} =$$
$$\left(Q(\Delta_k) - \sum_{n=2}^k C_n x^n(\Delta_k)h^{n-1}\right)\frac{1}{C_1}, \tag{11}$$

where $C_k = Q_k(\tau, \mu_0, \mu, \varphi)/Q_0(\tau, \mu_0, \mu, \varphi)$.

Equation (10) determines the analytic procedure for deconvolution of $x_{in}(\Delta)$ in accordance with Equation (1).

Figure 4 presents a set of multiple scattering signals describing electron energy loss in GO samples annealed at various temperatures.

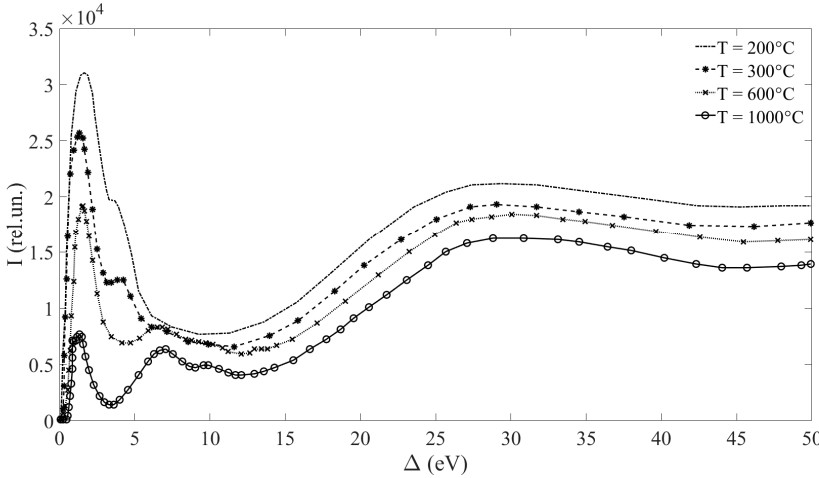

**Figure 4.** XPS spectra of thermally reduced graphene oxide after subtracting the peak formed by the C1 electrons not bonded chemically with oxygen ($\Delta = E_{C1s} - E$).

One can see the characteristic peaks related to two groups: 1—the peaks found in the electron-energy-loss region $\Delta < 5$ eV correspond to the electrons escaping from the C1 level. These electrons were not subjected to inelastic scattering but were emitted by atoms chemically bonded with oxygen. 2—the peaks found in the electron-energy-loss region $\Delta > 5$ eV correspond to the photoelectrons losing energy for excitation of plasmon oscillations. Note that even thermal processing at 1000 °C did not result in total removal of oxygen, and the peak of the relevant C-O bond is shown. The peak O-C-OH disappears at annealing at a temperature of 600 °C and higher. Chemically shifted C1 peaks should be subtracted from the experimental spectra presented in Figure 4. After that, the spectra obtained are treated using the above-described procedure (Equations (2)–(11)). The results of the treatment are shown in Figure 5.

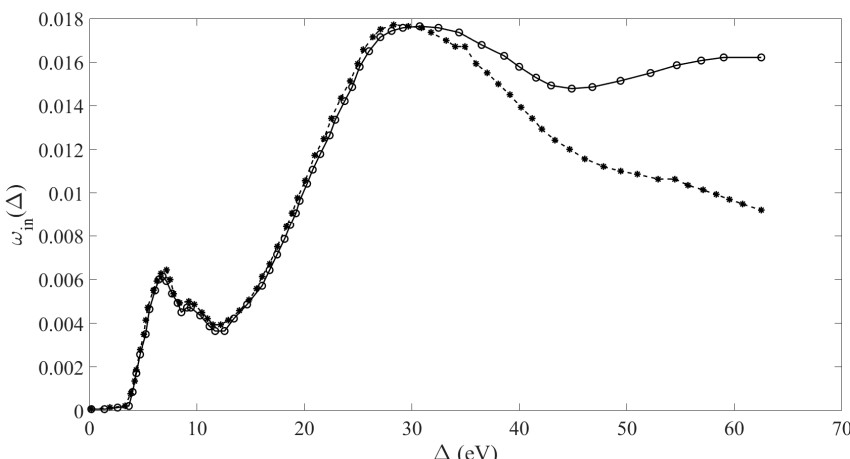

**Figure 5.** The $\omega_{in}(\Delta)$ describing the contribution of one-time energy loss processes in the spectrum. Circles correspond to the XPS spectrum of GO reduced at a temperature of 1000 °C, from which all the oxide peaks (CO, CO$_2$, and COH) have been extracted; asterisks—deconvolution spectra.

Figure 5 presents the $\omega_{in}(\Delta)$, describing the contribution of one-time energy loss processes into the spectrum. Three peaks are seen on these curves: the first one, at a resonant energy of approximately 7 eV, corresponds to $\pi$ plasmon oscillations; the second one, at an energy of approximately 10 eV, is hardly distinguishable; and the third one, at an energy of approximately 28 eV, corresponds to the $\pi + \sigma$ plasmon oscillations. The presented relationships are also inherent to pyrolytic graphite [9] (see also Figure 1).

The procedure for the reconvolution of the $\omega_{in}(\Delta)$ from PES spectra is a non-correct task of mathematical physics (ILL-POSED Problem). The best method for the solution of such tasks is trial and error [23].

For the solution of this task, it is necessary to subtract the elastic peak. This task consists of the calculation of spectra $Q_{fit}(\Delta, \mu_0, \mu, \varphi)$ with $x_{in}(\Delta)$, which contains a set of fitting parameters determined as a result of the trial-and-error procedure. This procedure is performed through the minimization of the function for the experimental spectra PES and REELS:

$$\gamma = \int_0^{\Delta_{max}} \left[ Q_{exp}(\Delta, \mu_0, \mu, \varphi) - Q_{fit}(\Delta, \mu_0, \mu, \varphi) \right] d\Delta, \tag{12}$$

where $Q_{exp}(\Delta, \mu_0, \mu, \varphi)$ is the experimental spectrum and $Q_{fit}(\Delta, \mu_0, \mu, \varphi)$ is the calculated spectrum, which is evaluated taking into account the hardware function of the energy analyzer, Doppler broadening $D(\Delta)$, the energy broadening of the probing electron beam and the energy spread of photoelectrons. The influence of between these features on the experimental conditions is given by the following relation:

$$Q_{fit}(\Delta, \mu_0, \mu, \varphi) = \int_0^{\Delta_{max}} Q(\Delta - \varepsilon, \mu_0, \mu, \varphi) D(\varepsilon) d\varepsilon. \tag{13}$$

For analytical calculation of functions $Q_k(\tau, \mu_0, \mu, \varphi)$, the reflection function partial coefficients $R_k(\tau, \mu_0, \mu, \varphi)$ are necessary. The set of matrix equations for the functions $Q_k(\tau, \mu_0, \mu, \varphi)$ and $R_k(\tau, \mu_0, \mu, \varphi)$ obtained on the basis of the invariant imbedding method are presented in [24]. The functions $Q_k(\tau, \mu_0, \mu, \varphi)$ are determined by the following equation:

$$\frac{\partial Q_k}{\partial \tau} + \frac{1}{\mu} Q_k - (1 - \delta_{k0}) \frac{1-\lambda}{\mu} Q_{k-1} = \delta_{k0} \lambda_\gamma f + \lambda_\gamma f \otimes R_k + \lambda Q_k \otimes x_{el}^+ + \lambda \sum_{j=0}^{k} Q_j \otimes x_{el}^- \otimes R_{k-j}, \tag{14}$$

where $x_{el}(\mu', \mu, \varphi') = \frac{\omega_{el}(\mu', \mu, \varphi')}{\sigma_{el}}$; $\omega_{el}(\mu', \mu, \varphi')$ and $\sigma_{el}$ are the differential and total elastic scattering cross sections, correspondingly; $x_{el}^+$ is the normalized differential inverse inelastic mean free path, which does not result in the transition of descending photoelectron

flow to ascending flow and vice versa; $x_{el}^-$ is the normalized differential inverse inelastic mean free path, which results in the transition of descending photoelectron flow to ascending flow and vice versa; $\lambda = \frac{\sigma_{el}}{\sigma_{el}+\sigma_{in}} = \frac{l_{in}}{l_{el}+l_{in}}$; $\lambda_\gamma = \frac{\sigma_\gamma}{\sigma_{el}+\sigma_{in}}$; $\sigma_\gamma$ is the total photoionization cross section; $f(\mu_0, \mu, \varphi) = \frac{1}{4\pi}\sum_{i=0}^{3} B_i P_i(\cos\psi)$; $P_i$ is the Legendre polynome; $\cos\psi = \mu_0\mu + \sqrt{(1-\mu_0^2)(1-\mu^2)}\cos\varphi$; $\psi$ is the scattering angle; $F(\mu_0, \mu, \varphi) = \sigma_\gamma f(\mu_0, \mu, \varphi)$ is the function of the photoelectron source or the photoionization cross section. The detailed description of the functions $f$, $B$ and $F$ can be found in [25,26].

The following designations are convenient to use: $x_{el}^+(\mu_0, \mu, \varphi) = x_{el}(\mu_0, \mu, \varphi)$; $\text{sign}(\mu_0, \mu) = 1$ is a part of the elastic scattering cross section which does not result in transformation of descending flow to ascending flow and vice versa (this part corresponds to a minor correction of the movement direction); $x_{el}^-(\mu_0, \mu, \varphi) = x_{el}(\pm\mu_0, \mp\mu, \varphi)$, $\text{sign}(\mu_0, \mu) = -1$ is a part of the elastic cross section corresponding to the reflection.

The following designation is used in Equation (14):

$$x_{el}^- \otimes R_m = \int_0^{2\pi}\int_0^1 x_{el}(\mu_0, \mu', \varphi') R_m(\tau, \mu', \mu'', \varphi'' - \varphi')\frac{d\mu'}{\mu'}d\varphi'. \tag{15}$$

The reflection of electrons is described by the function $R_k(\tau, \mu_0, \mu, \varphi)$, which is determined by the equation:

$$\frac{\partial R_k}{\partial \tau} + \left(\frac{1}{\mu_0} + \frac{1}{\mu}\right)R_k = \lambda x_{el}^-\delta_{k0} + \lambda x_{el}^+ \otimes R_k + \lambda R_k \otimes x_{el}^+ + \lambda R_0 \otimes x_{el}^- \otimes R_k +$$
$$\lambda R_k \otimes x_{el}^- \otimes R_0 + \lambda\sum_{j=1}^{k-1} R_j \otimes x_{el}^- \otimes R_{k-j} + (1-\delta_{k0})(1-\lambda)\left(\frac{1}{\mu_0} + \frac{1}{\mu}\right)R_{k-1}. \tag{16}$$

The numerical calculations were performed with the use of the electron elastic cross sections given in [27]. The IMFP was determined with the use of the TPP-2M formula [28,29]. The numerical matrix methods utilized for the solution of Equations (14) and (16) are described in detail in [24,30].

The main mechanisms of the electron energy loss in a solid are local energy loss for ionization and inter-band transitions, as well as non-local energy loss for excitation of plasmon oscillations. The corresponding $x_{in}(\Delta)$ is expressed in the following form:

$$x_{in}(\Delta) = \sum_{i=1}^{N_{pl}} \lambda_{pli} x_{pli}(\Delta) + \sum_{j=1}^{N_{ion}} \lambda_{ionj} x_{ionj}(\Delta), \tag{17}$$

where $x_{pli}(\Delta) = \frac{A_{pli}\Delta^\beta}{\left(\Delta^2 - \varepsilon_{pli}^2\right)^2 + \Delta^\alpha b_i^{4-\alpha}}$ is the normalized differential inverse inelastic mean free path for excitation of plasmon oscillation; $x_{ionj}(\Delta) = \frac{A_{ionj}}{\Delta^{2+a}}\eta(\Delta - J_{ionj})$ is the normalized differential inverse inelastic mean free path for ionization of the target atoms; $J_{ionj}$ is the ionization threshold; $\eta(\Delta - J_{ionj})$ is the Heaviside function; fitting parameters $\lambda_{pli}$, $\lambda_{ionj}$, $\alpha$, $\beta$, $b_i$ and $a$ should be determined in the course of the trial-and-error procedure; $A_{pli}$ and $A_{ionj}$ are evaluated using the normalization conditions:

$$\int_0^{E_0} x_{pli}(\Delta)d\Delta = 1, \tag{18}$$

$$\int_0^{E_0} x_{ionj}(\Delta)d\Delta = 1. \tag{19}$$

The following relation should be obeyed for fulfilling the normalization conditions (18) and (19):

$$\sum_{i=1}^{N_{pl}} \lambda_{pli} + \sum_{j=1}^{N_{ion}} \lambda_{ionj} = 1. \tag{20}$$

Figure 6 presents $x_{in}(\Delta)$, obtained on the basis of a repeated solution of the direct task in the case of a homogeneous target without taking into account differences in the mechanisms of energy loss in surface layers and in bulk remote from the surface.

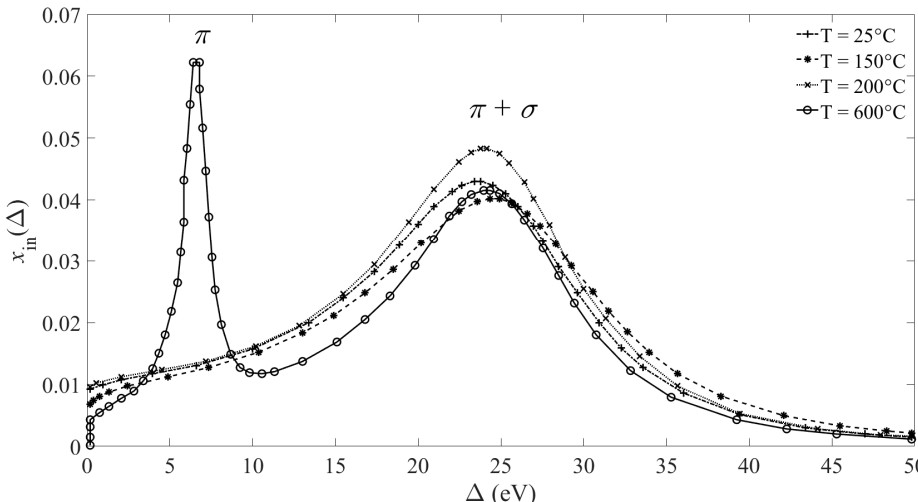

**Figure 6.** $x_{in}(\Delta)$ for GO annealed at different temperatures.

One should note that the cross sections presented in Figure 6, in contrast to Figure 5, were calculated without taking into account the broadening mechanisms related to the Doppler effect and the instrument function of the energy analyzer. This causes a distinction between the corresponding influences. The experimental data shown in Figure 6 are notably deformed because of the above-mentioned processes of signal broadening. The most comprehensive description of energy loss processes and the relevant information on the electron structure of reduced GO samples will be obtained as a result of the trial-and-error procedure taking into account differences in the energy loss processes in surface layers and homogeneous bulk structure remote from the surface. The detailed description of the $x_{inS}(\Delta)$ and $x_{inB}(\Delta)$ reconvolution procedure is given in [31] ($x_{inS}(\Delta)$ is the function of the energy loss in surface target layers; $x_{inB}(\Delta)$ is the function of the energy loss in target layers which are far from surface).

## 4. Discussion

The work describes the deconvolution of the differential inelastic scattering cross section $x_{in}(\Delta)$ from XPS spectra. The function $x_{in}(\Delta)$ or $\omega_{in}(\Delta)$ permits one to uniquely establish allotrope modification of the carbon lattice, where photoelectron movement occurs with electron energy loss for excitation of plasmon oscillations (Figure 1). Figures 5 and 6 present the function $x_{in}(\Delta)$, determined from the XPS signal related to multiple inelastic scattering. The data presented in Figure 5 have been obtained using a using a modification of the known Tougaard deconvolution method. This approach presents the solution of an inverse task, namely, the evaluation of the cross section from the electron energy loss spectrum. Figure 6 shows the function $x_{in}(\Delta)$ deconvoluted by means of calculating the direct scattering signal. Thereafter, the fitting parameters determining the function $x_{in}(\Delta)$ and matching the calculated and experimental spectra are selected. Notable differences in the deconvoluted $\omega_{in}(\Delta)$ on the basis of various approaches relate to the influence of Doppler line broadening, the energy analyzer and the energy spread of the X-ray probe. One should note that while the functions presented in Figures 5 and 6 differ in their shape, the energy positions of $\pi$ and $\pi + \sigma$ plasmon peaks correspond to those inherent in pyrolytic graphite. This indicates that annealing promotes the transition of the system into a minimum potential energy state, which is a pyrolytic graphite. All the characteristics of carbon samples presented in Figure 6 show greater differences; however, the data treatment reflected in Figure 5 and described by Equations (3)–(12) is much less laborious than the realization of the fitting process resulting in Figure 6.

Note that, in some cases, a simultaneous analysis of samples by means of REELS and XPS spectroscopy is possible. It has been demonstrated that the cross sections deconvo-

luted from XPS by the usage of the approach described in the present work correspond qualitatively to REELS spectra and vice versa [32].

The main goal of the present work is the demonstration of the possibilities of the approach including the analysis of the electron-energy-loss region adjacent to the peak formed by the electrons escaping in vacuum without energy loss. Now the most known method is peak shape analysis, as described in [6]. The present work demonstrates the possibilities of the analysis. For its realization, it is necessary to cover the electron-energy-loss region 50–100 eV at a high resolution. The experimental spectra shown in the present article cover the electron-energy-loss region 50–60 eV. One exception is Figure 3, where the electron-energy-loss region slightly exceeding 10 eV is covered. This spectrum contains only the $\pi$ plasmon peak, which is kept constant; however, the $\pi + \sigma$ plasmon peak that would permit one to obtain more detailed information on electronic excitations of the sample has not recorded.

The study of GO samples reduced at different temperatures indicated the presence of C–O bonds at an annealing temperature of 1000 °C (see Figure 3). The deciding influence of oxidation on the electronic properties of samples is manifested at annealing temperatures below 200 °C. In this case, the $\pi$ plasmon peak is absent in spectra. At higher annealing temperatures, $\pi$ plasmon effect on the $\omega_{in}(\Delta)$ is observable. This contribution increases as the annealing temperature is increased to approximately 600 °C. A further increase in the annealing temperature to 1000 °C does not significantly change the $\omega_{in}(\Delta)$ (see Figure 4). At annealing temperatures exceeding 600 °C, the $\omega_{in}(\Delta)$ is close to that for pyrolytic graphite, for which XPS spectra are presented in [33]. Note that the characteristics determined here on the basis of XPS spectra relate to the surface layer of a sample at approximately the nanometer scale, which corresponds to the mean free path for inelastic scattering.

In contrast to GO, the oxidation of CNT does not influence the electron structure of samples and does not change the characteristic energy loss. In other words, the honeycomb structure of CNTs is not affected by additional chemical bonds. Only chemical shifts in the peaks formed by the electrons escaping in vacuum without inelastic energy losses are observable.

Three approaches have been presented for PES analysis of XPS spectra for the determination of the properties of a sample from the $\omega_{in}(\Delta)$. The first two methods are the most useful for technological applications due to the ease of realization and enabling the determination of the real $x_{in}(\Delta)$, respectively.

One should note that from the PES spectrum in Figure 2 shows that both the position and the shape of $\pi$ plasmon peak do not change in the course of the oxidation of nanotubes. In this case, the $\omega_{in}(\Delta)$ deconvolution procedure is not required.

**Author Contributions:** Conceptualization, V.P.A. and A.V.E.; methodology, V.P.A. and L.G.L.; software, L.G.L.; formal analysis, V.P.A.; investigation, A.V.E., K.I.M. and S.V.S.; data curation, G.S.B., K.I.M., S.V.S. and L.G.L.; writing—original draft preparation, V.P.A. and A.V.E.; writing—review and editing, V.P.A. and A.V.E.; All authors have read and agreed to the published version of the manuscript.

**Funding:** This study was financially supported by the Ministry of Science and Higher Education of the Russian Federation (project No. FSWF-2023-0016).

**Data Availability Statement:** All information and data are in this article and references.

**Conflicts of Interest:** The authors declare no conflict of interest.

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
