# Peer review of "Comparative Investigation of XPS Spectra of Oxidated Carbon Nanotubes and Graphene"

_biophysica, doi:10.3390/biophysica3020020_

Round 1

Reviewer 1 Report

Article – “Comparative investigation of XPS spectra of oxidated carbon nanotubes and graphene”

Authors – Viktor P. Afanas’ev, Grigorii S. Bocharov, Alexander V. Eletskii, Lidiya G. Lobanova, Konstantin I. Maslakov and Serguei V. Savilov

Summary – The authors study the XPS spectra of thermally reduced graphene oxide and chemically oxidized carbon nanotubes. The comparison inferred that the electronic structure varies in graphene oxide depending on the degree of oxidation. However, oxidation degree does not alter the electronic structure of CNT. The authors further derive an equation for normalized differential inverse inelastic mean free path (nDIIMFP). Plotting the data in terms of nDIIMFP vs binding energy can give a direct visualization of the electron energy loss during XPS measurement.

Overall, the work presented shows an important method of analysis to remove the spectral broadening caused by instrumental or experimental interferences. This manuscript can be accepted after the following comments are addressed by the authors.

Major Comments

1.     Please improve the grammar of the manuscript for the readers to understand the science better, especially in the introduction section.

2.     Consider modifying the title, since the major paper talks about the importance of nDIIMFP rather than comparing CNT and graphene XPS data.

3.     Why is the CNT XPS spectrum not plotted with the nDIIMFP -  vs  ? Comparing this plot with Figure 6 (graphene) would be a good addition, and important for the title of the article.

4.     Please improve the figure captions to include specific details of the data represented in them.

Minor Comments

1.     Line 24. Please cite references for the sentence- “Oxidation of carbon nanomaterials changes considerably their electron properties.”

2.     Line 29. Please cite references for the statement – “ ….(XPS) which does not inflict damages to the material.”

3.     Line 81. Please add the duration for which the samples were annealed.

4.     Line 89. Please add information about the source of carbon nanotubes.

5.     In Figure 3, please discuss the rise of a shoulder around 288 eV on increasing oxidation of CNT.

6.     In Figure 5, kindly mention the specific dataset used to obtain the plot. What temperature was the GO annealed at?

Reviewer 2 Report

In their submitted manuscript, Afanas’ev et al. present XPS experiments on graphene and CNTs after oxidation by acid and thermal desoxygenation. A detailed analysis regarding the inelastic mean free path and electron energy loss phenomena constitutes the main body of this work.

In general, this is a good piece of work, but I wonder why the authors submitted this to “Biophysica”, it seems to be the wrong journal from my perspective. Among the MDPI journals, “Solids”, “C”, “Surfaces” or the “Spectroscopy Journal” would be more fitting for this paper.

Comments regarding the content of the manuscript can be found in the attached PDF.

Round 2

Reviewer 2 Report

The manuscript has been improved substantially and can now be considered publishable.

As written in my previous review, I still think that among MDPI journals, “Solids”, “C”, “Surfaces” or the “Spectroscopy Journal” would be thematically more fitting than "Biophysica", but if the editorial office thinks this fits, I will not block it.